# Characterization of the Outer Membrane Vesicles of *Pseudomonas aeruginosa* Exhibiting Growth Inhibition against *Acinetobacter baumannii*

**DOI:** 10.3390/biomedicines12030556

**Published:** 2024-03-01

**Authors:** Jin-Woong Suh, Jae-Seong Kang, Jeong-Yeon Kim, Sun-Bean Kim, Young-Kyung Yoon, Jang-Wook Sohn

**Affiliations:** 1Division of Infectious Diseases, Department of Internal Medicine, Korea University College of Medicine, 73, Inchon-ro, Seongbuk-gu, Seoul 02841, Republic of Korea; sunthes@naver.com (J.-W.S.); young7912@korea.ac.kr (Y.-K.Y.); 2Institute of Emerging Infectious Diseases, Korea University, 145, Anam-ro, Seongbuk-gu, Seoul 02841, Republic of Korea; naya3482@hanmail.net

**Keywords:** *Pseudomonas aeruginosa*, *Acinetobacter baumannii*, Gram-negative bacteria, outer membrane vesicles, proteomics

## Abstract

We investigated the *Pseudomonas aeruginosa* (PA) outer membrane vesicles (OMVs) and their effect on *Acinetobacter baumannii* (AB) growth in vitro. The inhibitory effects of PA on AB were assessed using a cross-streak assay. The OMVs were extracted through high-speed centrifugation, tangential flow filtration, and ultracentrifugation and characterized by sodium dodecyl sulfate-polyacrylamide gel electrophoresis (SDS-PAGE), transmission electron microscopy (TEM), and nanoparticle tracking assays (NTAs). Proteomic analysis was conducted to compare the OMVs of different PA strains. PA022 exhibited more pronounced inhibition of AB growth compared with PA ATCC 27853. TEM confirmed the presence of OMVs in both PA022 and PA ATCC 27853, revealing phospholipid bilayer structures. The NTA revealed similar sizes and concentrations. Proteomic analysis identified 623 and 538 proteins in PA022 and PA ATCC 27853 OMVs, respectively, with significant proportions of the outer membrane and extracellular proteins, respectively. Importantly, PA022 OMVs contained six known virulence factors and motility-associated proteins. This study revealed the unique characteristics of PA OMVs and their inhibitory effects on AB growth, shedding light on their role in bacterial interactions. Proteomic analysis provides valuable insights into potential pathogenic functions and therapeutic applications against bacterial infections.

## 1. Introduction

Outer membrane vesicles (OMVs) are spherical structures originating from the outer membrane of bacterial cells [1]. Since the discovery of Gram-negative bacterial (GNB) OMVs in the 1960s, they have garnered interest due to their role in regulating immunomodulatory activity and biochemical molecule transport [2]. OMVs have crucial biological functions that are associated with pathogenesis and competition with other organisms [3]. Additionally, OMVs may target and eliminate neighboring competing bacteria by encapsulating toxic molecules, such as antimicrobial peptides and enzymes [4]. Understanding OMV release and its involvement in bacterial competition can provide insight into microbial competition and pathogenesis [5]. Furthermore, the application of proteomic analysis to examine bacterial OMVs is valuable for understanding bacterial interspecific interactions. Comparative analyses of OMV proteomes released by different bacteria have shed light on the diverse strategies employed by various species and strains to compete or thrive in their environments [6].

Hospital-acquired infections (HAIs) caused by multidrug-resistant Gram-negative bacteria (MDR-GNBs) pose significant challenges in the treatment of critically ill patients [7]. Among these, *A. baumannii* has emerged as a serious nosocomial pathogen and has become a global concern. The 2019 Antibiotic Resistance Threats Report of the Centers for Disease Control and Prevention categorizes carbapenem-resistant *A. baumannii* (CRAB) as a pathogen requiring immediate attention [8]. *P. aeruginosa* secretes various virulence factors that are crucial for mediating its interactions with other bacterial species, leading to infection and increased virulence [9]. Additionally, *P. aeruginosa* produces antibacterial factors that can be delivered via the OMVs to eliminate competing bacteria [10]. Notably, the type VI secretion system is used by bacteria to inject toxins and other effector proteins into neighboring bacterial cells and the host [11]. Analysis of OMVs derived from *P. aeruginosa* has the potential to identify novel antibacterial agents. However, the precise mechanisms and functions of OMVs in bacterial interactions remain unclear. Understanding OMVs that inhibit bacteria is crucial given the diverse OMV proteomics resulting from variations in extracellular vesicle biogenesis [12]. Therefore, this study aimed to investigate the interactions between *P. aeruginosa* and *A. baumannii* and determine the microbiological characteristics of *P. aeruginosa* OMVs against *A. baumannii* in vitro.

## 2. Materials and Methods

### 2.1. Sample Selection of GNBs

To ensure a comprehensive and unbiased representation of isolates, *P. aeruginosa* and *A. baumannii* were selected from a de-identified bacterial collection at a tertiary care hospital. Among these strains, *P. aeruginosa* was independently isolated from aseptic blood samples (*P. aeruginosa* 068 [PA068]), sputum (PA008) obtained from a patient with pneumonia, and urine (PA022) collected from a patient with urinary tract infection admitted in the intensive care unit (ICU). The standard laboratory strain *P. aeruginosa* ATCC 27853 (PA ATCC 27853), isolated from blood, was used as a control. *A. baumannii* was isolated from the environment [*A. baumannii* MIC34 (AB MIC34)], and blood samples (AB004, AB037, AB SSH20, and AB KBU02) collected in the ICU. All tested strains were revived from frozen stocks stored at −80 °C by streaking them on blood agar plates and incubating them overnight. The bacterial colonies were prepared using a modified cross-streak assay. The identification and antibiotic susceptibility testing of the strains were performed using a VITEK II (BioMérieux, Hazelwood, MO, USA) and a MicroScan WalkAway-96 Plus system (Beckman Coulter, Inc., Brea, CA, USA) (Table 1). The antimicrobial susceptibility of the tested strains was interpreted according to the Clinical Laboratory Standards Institute (CLSI) reference guidelines [13]. This study was approved by the Institutional Review Board of the Korea University Anam Hospital (IRB number: 2022AN0238).

### 2.2. Modified Cross-Streak Assay

The interaction between *P. aeruginosa* and *A. baumannii* was assessed using a modified cross-streak assay, as outlined previously [14] (Appendix A). To investigate the inhibitory effects of *P. aeruginosa* on *A. baumannii*, cross-streak assays were performed on Mueller–Hinton agar (MHA) plates using all strains of *P. aeruginosa* and *A. baumannii*. A sterile cotton swab was used to draw a horizontal line of *P. aeruginosa* colonies (1 × 10^8^ CFU/mL = 0.5, McFarland [McF] standard concentration) across the plate, ensuring no direct contact between the bacteria. Simultaneously, *A. baumannii* colonies (0.5 McF) were vertically inoculated across the plates using a cotton swab. Subsequently, all plates were air-dried for 15 min and then incubated. After 24 h, a transilluminator (FUSION Solo 7S; VILVER, Collegien, France) was used for digital photography to visualize the growth of the tested bacterial strains on the plates.

### 2.3. Preparation and Quantification of OMVs

*P. aeruginosa*-derived OMVs were prepared according to a previously described method with minor modifications [15]. As no protocol has been established for the preparation of OMVs, the presence of free soluble extracellular proteins generated during OMV extraction was minimized by high-speed centrifugation, concentration, and ultracentrifugation. The *P. aeruginosa* strains (PA022 and PA ATCC 27853) were cultured in 4 L of MH culture broth at 37 °C for 10 h in a shaking incubator. Subsequently, bacterial cells were removed by high-speed centrifugation using Supra R22 (Hanil Scientific Inc., Gimpo, Republic of Korea) at 6000× *g* for 1 h at 4 °C. The resulting supernatant was filtered through a 0.22 μm bottle top filter membrane (Merck KGaA, Darmstadt, Germany) to remove bacterial debris. The resulting filtrate containing OMVs was concentrated using a tangential flow filtration system with a VivaFlow 50R module (Sartorius, Goettingen, Germany) and a 5 kDa cut-off membrane. The concentrated OMVs were pelleted down using Optima L-100 XP Ultracentrifuge (Beckman Coulter, Brea, CA, USA) at approximately 200,000× *g* for 2 h at 4 °C using a type 70 Ti rotor (Beckman Coulter, Brea, CA, USA) in polypropylene centrifuge tubes (Beckman Coulter, Brea, CA, USA). Pellets containing OMVs were obtained and resuspended in Dulbecco’s phosphate-buffered saline (DPBS). Each sterilized OMV suspension was spread onto a blood agar plate and incubated to confirm the absence of contamination. The prepared OMVs were divided into 500 μL with DPBS and then stored at −80 °C for further use. The protein concentrations of OMVs from PA ATCC 27853 and PA022 were determined using the Bradford protein assay (Bio-Rad, Hercules, CA, USA) according to the manufacturer’s instructions. Each OMV was quantified using SpectraMax i3x (Molecular Devices, LLC., San Jose, CA, USA). The Bradford protein assay for each OMV was performed in triplicate.

### 2.4. Transmission Electron Microscope

OMV extraction was confirmed by transmission electron microscopy (TEM). Following the preparation of PA ATCC 27583 and PA022 OMVs, they were directly loaded onto 0.5% formvar-coated 300-mesh copper grids and negatively stained with 2% uranyl acetate, following established protocols [16]. Subsequently, the samples were washed with PBS and visualized using a TEM H-7650 (Hitachi, Tokyo, Japan) operating at 80 kV.

### 2.5. Sodium Dodecyl Sulfate-Polyacrylamide Gel Electrophoresis

To perform sodium dodecyl sulfate-polyacrylamide gel electrophoresis (SDS-PAGE), 10 μL bacterial cells from PA ATCC 27853 and PA022 were utilized. The samples of two OMVs and two bacterial total proteins were boiled at 100 °C in a heat block for 10 min. Pre-stained protein markers (Bio-Rad, Hercules, CA, USA) were used to determine the protein size. Four samples were loaded onto 4–15% SDS-PAGE gels and subjected to electrophoresis. Subsequently, Coomassie blue was used for visualization.

### 2.6. Nanoparticle Tracking Analysis of OMVs

The quantity and size of OMVs of *P. aeruginosa* strains were determined using NanoSight NS300 (Malvern Instruments Ltd., Malvern, UK). The OMV samples were thawed at room temperature and diluted 1/100 before analysis. The samples were infused using a NanoSight NS300 syringe pump. The concentration (particles/mL), mean size (nm), and mode size (nm) were determined, and the average of three readings was calculated and plotted as the particle size versus the number of particles/mL. The measurements were performed at an ambient room temperature (range: 22.5–22.6 °C). Five readings, each lasting 60 s, were taken in instrument-optimized settings with the options set to “automatic” and viscosity set to “water” (0.939–0.941 cP). An automated image set-up (camera level and focus) was selected whenever available for video enhancement. A total of 607 frames per sample were analyzed using the NTA software version 3.4 (Malvern Instruments Ltd., Malvern, UK), with a detection threshold of five (in arbitrary units) [17].

### 2.7. Modified Time-Kill Assay

Modified time-kill assays were performed using OMVs of *P. aeruginosa* PA ATCC 27853 and PA022 against *A. baumannii*. The final OMV concentration of each *P. aeruginosa* was set to 40 µg/mL, as previously described with some modifications [18]. The final tubes contained MH broth supplemented with OMVs of each *P. aeruginosa* strain and 1 × 10^5^ CFU/mL of *A. baumannii* (AB037) and were incubated for 10 h. Serial dilutions were performed at 0, 2, 4, 6, and 10 h. The diluted samples were plated on MHA plates in triplicate, and the average CFU/mL was determined. The results of the time-kill curves were reported as log CFU/mL. The log reduction in CFU/mL was determined based on the control using the following equation: log (control)-log (PA022 or PA ATCC 27853). Bactericidal and bacteriostatic effects were defined as a reduction in the baseline log CFU/mL by >3 log CFU/mL, with the log CFU/mL remaining approximately the same as the baseline log CFU/mL concentration over time. The synergistic effect was defined as a ≥2 log CFU/mL reduction in colony counts compared with the most active control after a 24 h incubation.

### 2.8. Proteomics

#### 2.8.1. Proteomic Sample Preparation

Gels were washed with 5% acetonitrile (ACN) (Sigma-Aldrich, St. Louis, MO, USA) in 25 mM NH_4_HCO_3_ (Sigma-Aldrich, St. Louis, MO, USA). After cutting the gels into small pieces, they were destained twice with 50% ACN in 25 mM NH_4_HCO_3_, dehydrated with ACN, and dried. Proteins were enzymatically digested overnight at 37 °C using trypsin (Promega, Madison, WI, USA) in 40 mM NH_4_HCO_3_. The resulting tryptic peptides were extracted twice with 0.5% trifluoroacetic acid (TFA) in 50% ACN, followed by a final extraction with ACN. The extracted peptides were subsequently dried and desalted using C18 spin columns (Harvard Apparatus, Holliston, MA, USA).

#### 2.8.2. Mass Spectrometry Analysis

Liquid chromatography–mass spectrometry (LC-MS) analysis was performed using a NanoElute LC system (Bruker Daltonics, Bremen, Germany) connected to a timsTOF Pro (Bruker Daltonics) using a CaptiveSpray nanoelectrospray ion source (Bruker Daltonics). Approximately 200 ng of the peptide digest was injected into a capillary C18 column (IonOpticks, Fitzroy, VIC, USA), and gradient elution was performed using 0.1% formic acid in water (A) and 0.1% formic acid in ACN (B) as the mobile phase. Mass spectral data ranging from *m*/*z* 100 to 1700 were acquired in the parallel accumulation and serial fragmentation (PASEF) mode [19]. Ion mobility resolution was set to 0.60–1.60 V·s/cm over a ramp time of 100 ms, with collisional energy ramped stepwise. Ten PASEF MS/MS scans per cycle were used for the data-dependent acquisition.

#### 2.8.3. Processing, Quantification, and Statistical Analysis of MS Data

The raw files were processed using Peaks software (version 10.5) and matched to tryptic peptide fragments from the UniProt Pseudomonas protein database, allowing a maximum of three missed cleavages [20]. Mass tolerance was set at 15 ppm and 0.05 Da for precursor and fragment mass, respectively. Variable modifications included methionine oxidation and N-terminal acetylation. The false discovery rate (peptide) was set at 0.01 for peptide identification. OMV proteins from *P. aeruginosa* identified by LC-MS were annotated for subcellular localization using pSORTb version 3.0.3 [21].

### 2.9. Statistical Analysis

The quantification data were expressed as the mean ± standard deviation (SD). Statistical significance was determined using Student’s *t*-test. A *p* value of <0.05 was considered significant. IBM SPSS Statistics for Windows (version 23.0; IBM Corp., Armonk, NY, USA) was used to perform all statistical analyses.

## 3. Results

### 3.1. Interaction of GNBs with P. aeruginosa in Cross-Streak Assay

In the cross-streak assay, *P. aeruginosa* (PA068, PA008, and PA ATCC 27853) showed no inhibitory effects on *A. baumannii*. However, PA022 showed significant growth inhibitory zones for all tested *A. baumannii* adjacent to *P. aeruginosa* after 72 h (Figure 1). Therefore, PA022 had an inhibitory effect on *A. baumannii*.

### 3.2. OMV Production of PA022 Compared with PA ATCC 25783

According to a cross-streak assay, PA022 inhibited the growth of *A. baumannii*. Therefore, it was selected for the OMV extraction. To maximize OMV harvesting, OMV extraction was performed after incubation for 10 h using a growth curve. To establish OMV yields and accurately standardize assays, quantification of extracted OMVs is essential. TEM analysis confirmed the presence of spherical phospholipid bilayer structures in PA022 OMVs and PA ATCC 27853 OMVs (Figure 2A,B). SDS-PAGE analysis indicated distinct electrophoretic patterns of OMVs from PA ATCC 27853 and PA 022, strongly suggesting variations in the protein components of each OMV (Figure 2C). The PA 022 OMVs exhibited a prominent band at a molecular weight of 50 kDa in the middle lane. Nanoparticle tracking analysis (NTA) revealed OMVs of similar sizes and concentrations in PA022 and PA ATCC 27853 cells (Figure 2D). PA 022 OMVs showed a mean concentration of 5.07 × 10^8^ [±standard deviation (SD), 5.97 × 10^7^] particles/mL, a mean size of 184.8 nm (±SD, 5.3 nm), and a mode size of 173.1 nm (±SD, 3.5 nm). However, PA ATCC 27853 OMVs showed a mean concentration of 5.99 × 10^8^ (±SD, 9.38 × 10^7^) particles/mL, with a mean size of 190.3 nm (±SD, 1.8 nm) and a mode size of 177.3 nm (±SD, 2.7 nm).

### 3.3. Modified Time-Kill Assay

Upon exposure to OMVs of PA022 and PA ATCC 27853 for 6 h, *A. baumannii* exhibited growth inhibition compared with the observed effect in the control group. Specifically, PA022 OMVs displayed significant inhibitory activity compared with PA ATCC 27853 OMVs at 6 h (*p* < 0.01) and 10 h (*p =* 0.03) of incubation (Figure 3). Although no significant decrease in *A. baumannii* CFUs was observed with either PA022 or PA ATCC 27853 OMVs until 4 h, PA022 OMVs induced significant log reductions in the colony counts of *A. baumannii* at 6 (*p* = 0.02) and 10 h (*p* < 0.01).

### 3.4. OMV Proteomics

The proteins present in OMVs from *P. aeruginosa* strains were identified through LC-MS/MS, revealing 623 and 538 proteins in PA022 and PA ATCC 27853, respectively (Figure 4A). All proteomic datasets were deposited in the Proteomics Identification Database (https://www.ebi.ac.uk/pride/archive/projects/PXD045094, accessed on 18 September 2023). The proteomic data of PA022 and PA ATCC 27853 OMVs are shown in Appendix A. Subsequent analysis focused on the subcellular localization of each *P. aeruginosa* OMV proteome (Figure 4B, Appendix A). Periplasmic proteins, mainly related to transport, were exclusively detected in PA022 OMVs (2.9%). The most abundant proteins with known functions localized in the cytoplasmic membrane were the membrane transport protein and signal peptidase in PA ATCC 27853 OMVs, and the Sec translocon accessory complex subunit YajC and ATP synthase subunit C in PA022 OMVs. The ribosomal proteins identified were the 50S ribosomal protein L4 in ATCC 27853 OMVs and the 30S ribosomal protein S20 in PA022 OMVs. The RplQ protein, also known as 50S ribosomal protein L17, was found in *P. aeruginosa* strains. Proteins with unknown subcellular localization sites were also detected in PA022 (18.8%) and PA ATCC 27853 (5.2%).

The outer membrane and extracellular proteins constituted 34.5% and 14.9% of all proteins identified in PA022 and PA ATCC 27853 OMVs, respectively (Appendix A). Outer membrane proteins are primarily involved in membrane biogenesis and transport. The pathogenic characteristics of OMVs are assumed to depend primarily on the actions of the outer membrane and extracellular proteins [22]. Results of the additional analyses of the outer membrane and extracellular proteins of PA022 and PA ATCC 27853 OMVs are shown in Table 1. Only one extracellular protein-associated motility factor was detected in PA ATCC 27853 OMVs, whereas six extracellular proteins in PA022 OMVs are known to act as virulence proteins, including elastase and pseudolysin, and motility proteins, such as pilin, flagellin, and phage tail protein (Table 1). The pathogenic characteristics of OMVs are primarily influenced by the actions of the OMV proteins [20]. To determine the function of OMVs, the identified proteins were analyzed using the Clusters of Orthologous Groups database [23]. The PA022 OMV proteins were predominantly associated with virulence, cell motility, and post-translational modifications (Figure 4C).

## 4. Discussion

Our study is the first to investigate the inhibitory effects of *P. aeruginosa* OMVs on the growth of *A. baumannii*. Our results revealed that *P. aeruginosa* OMVs inhibit *A. baumannii* growth, suggesting the involvement of secreted proteins in this inhibitory process. These findings are significant in the context of previous studies reporting the inhibitory effects of small molecules or metabolic pathways produced by *P. aeruginosa* on the growth of *A. baumannii* [24] and respiratory pathogens [25]. Furthermore, our study found that OMVs from *P. aeruginosa* inhibited *A. baumannii* growth, as evidenced by cross-streak assays and supported by a time-kill assay. Nevertheless, a previous study did not observe the antibacterial effects of *P. aeruginosa* extracellular compounds on *A. baumannii* during in vitro co-culture [26]. This discrepancy underscores the potential variations in OMV-mediated pathogenicity owing to the differences in bacterial species, physical properties, and environmental conditions.

Both the cross-streak assay and the time-kill assay are well-known methods for evaluating antimicrobial activity in vitro. The cross-streak assay is a rapid screening method for assessing microorganism antagonism, while the time-kill assay is recognized as the most appropriate method for determining bactericidal effects [27]. Our study revealed a significant inhibition zone of *A. baumannii* by PA022 observed after 72 h in the cross-streak assay, whereas PA ATCC 27853 formed distinct colonies overlapping with *A. baumannii*. Furthermore, the time-kill assay revealed significant growth inhibition of *A. baumannii* by extracted PA022 OMVs after 6 h, and PA ATCC 27853 also exhibited growth inhibition effects compared to the control group after 10 h. These findings suggest that the inhibition phenomenon formed distinct colony-like boundaries in the cross-streak assay.

The extracted OMVs were confirmed by SDS-PAGE, TEM, and NTA. Based on the SDS-PAGE results, the clinical PA 022 strain exhibited an OMV band pattern closely resembling that of previously reported *P. aeruginosa* OMVs [28]. Size distribution analysis of OMVs produced by both *P. aeruginosa* strains using NTA revealed that OMVs produced through various biogenesis mechanisms had sizes ranging from approximately 50 to 400 nm at similar concentrations (Figure 2D). These findings are consistent with those of a previous study reporting a *P. aeruginosa* OMV size distribution within 400 nm [28,29]. Previous studies have indicated that mutations in specific membrane protein genes can alter the outer membrane structure of GNBs, resulting in the hypervesiculation of OMV production [30]. Additionally, variations in vesicle morphology, depending on the strain of origin, contribute to the differences in OMV composition [31]. Therefore, the observed differences in OMVs among *P. aeruginosa* strains may be attributed to certain factors such as alterations in membrane structure and vesicle formation.

Research on various extracellular materials beyond OMVs, including free soluble extracellular proteins [15] and outer-inner membrane vesicles (O-IMVs) [32], has also been reported. These materials exhibit different characteristics. Free soluble extracellular proteins are unenclosed proteins released into the extracellular space, whereas both O-IMVs and OMVs are membrane-bound vesicles. O-IMVs contain components of both outer and inner membranes, while OMVs are primarily composed of an outer membrane. Each of these components plays a distinct role in bacterial physiology and interactions with their surroundings [3,15,32] despite the evident significance of these distinctions. However, it is essential to acknowledge these distinctions using TEM, NTA, SDS-PAGE, and proteomic analyses. Based on these experimental methods, our findings align more closely with OMVs and are distinct from previously reported extracellular materials [33]. Further studies are required to differentiate and characterize these components better.

A total of 161 proteins in PA 022 were identical to those in PA ATCC 27853. The proteins identified in our study were consistent with those previously reported for *P. aeruginosa* [34]. The octatricopeptide repeat (Opr) family was identified as the most abundant outer membrane protein in *P. aeruginosa*. Among the Opr proteins, OprF, a major outer membrane protein, has been investigated for its role in antimicrobial drug resistance [35]. Additionally, OprG [36], a known virulence factor, and OprD [37] and OprO [38], which are associated with antibiotic resistance, were found in both PA022 and PA ATCC 27853. GroEL was also identified as an abundant protein in both *P. aeruginosa*. GroEL plays a pivotal role in the response to heat shock, thereby enhancing bacterial adaptation to various environments. Furthermore, they can stimulate innate immune responses against clinically significant bacterial pathogens [39]. All of these proteins have been previously identified as *P. aeruginosa* OMV proteins [40,41,42].

The pathogenic characteristics of OMVs depend on the actions of the outer membrane and extracellular proteins [22]. A total of 215 and 80 outer membrane and extracellular proteins were identified in PA022 and PA ATCC 27853, respectively (Table 1). Notably, several extracellular membrane proteins associated with host–bacterial interactions are recognized as virulence factors in clinical *P. aeruginosa*. PagL [22] is implicated in environmental adaptation, whereas FliC induces cytokine release by macrophages treated with *P. aeruginosa* OMVs [43]. Furthermore, several outer membrane proteins associated with antibiotic efflux have been identified, including OprM [44], OprJ [45], and opmH [46]. Additionally, LasB [47], SylB [48], HR1 [49], and OprC [49] were associated with virulent characteristics of invasive *P. aeruginosa*. The PilA protein, a key component of type IV pili, plays a critical role in the virulence of *P. aeruginosa* [50]. Our findings, along with those of previous studies, revealed that clinical *P. aeruginosa* OMVs, which inhibit *A. baumannii* growth, contain diverse virulence factors compared with the standard strain. These virulence-associated proteins have the potential to trigger outer membrane proteins involved in antibiotic resistance and host cell invasion through proteolytic processes. However, the specific mechanisms of virulence-related proteins involved in bacterial interactions remain unclear. Therefore, further investigations are required to elucidate the precise mechanisms governing the interplay between these virulence-related proteins and bacterial interactions.

Proteomic analysis of *P. aeruginosa* OMVs revealed 14 orthologous clusters. Notably, elastase, a protease known for its role in degrading host proteins and enhancing bacterial virulence [51], was identified in the OMVs from PA022. However, the effects of virulence factors on bacterial growth can vary depending on the bacterial species and the environment [52]. Further research is required to better understand the relationship between proteases and bacterial growth. Motile proteins are involved in interactions between microorganisms, particularly during infectious pathogenic processes [53]. In particular, *P. aeruginosa* utilizes type IV pili for interspecies interaction [54]. Post-translational modifications (PTMs) in *P. aeruginosa* can affect bacterial competition by modulating surface-associated proteins [55], secretion systems [56], and quorum-sensing regulators [57]. Our study showed that PA022 had a higher abundance of proteins associated with motility and PTMs. Our findings highlight the challenge of identifying pathogenic OMV proteins and conducting large-scale comparative studies owing to their diversity. Future research should focus on identifying antibacterial proteins based on their functions, using improved proteomic data.

Our study has several limitations. First, the study primarily focused on characterizing OMVs and their potential inhibitory effects in vitro, which may not represent the complex dynamics and interactions between bacteria in clinical settings. Second, the small sample size limits the statistical power and generalizability of the results. Further investigations using larger samples are required to validate the precise molecular interactions and signaling pathways involved in OMV-mediated inhibitory interaction.

## 5. Conclusions

The study explored the interaction between *P. aeruginosa* and *A. baumannii*, unveiling the potential of *P. aeruginosa* to inhibit the growth of *A. baumannii* in vitro. Notably, our results suggest that OMV proteins from *P. aeruginosa* (PA022) could inhibit the growth of *A. baumannii* strains, offering promising avenues for the development of novel therapeutic strategies. The proteins within PA022 OMVs may serve as viable alternatives to manage and mitigate infections caused by *A. baumannii*. Furthermore, the proteome profiles of the outer membrane and extracellular proteins in PA022 OMVs, which include proteins associated with virulence, motility, and PTM, provide a basis for future research aimed at elucidating the pathogenic mechanisms of *P. aeruginosa* OMVs. These findings not only enrich our understanding of microbial interactions but also open the door to novel approaches against bacterial infection. Moving forward, further research using advanced methodologies may be able to precisely identify the OMV proteins of *P. aeruginosa* that induce the growth inhibition of *A. baumannii*.

## Figures and Tables

**Figure 1 biomedicines-12-00556-f001:**
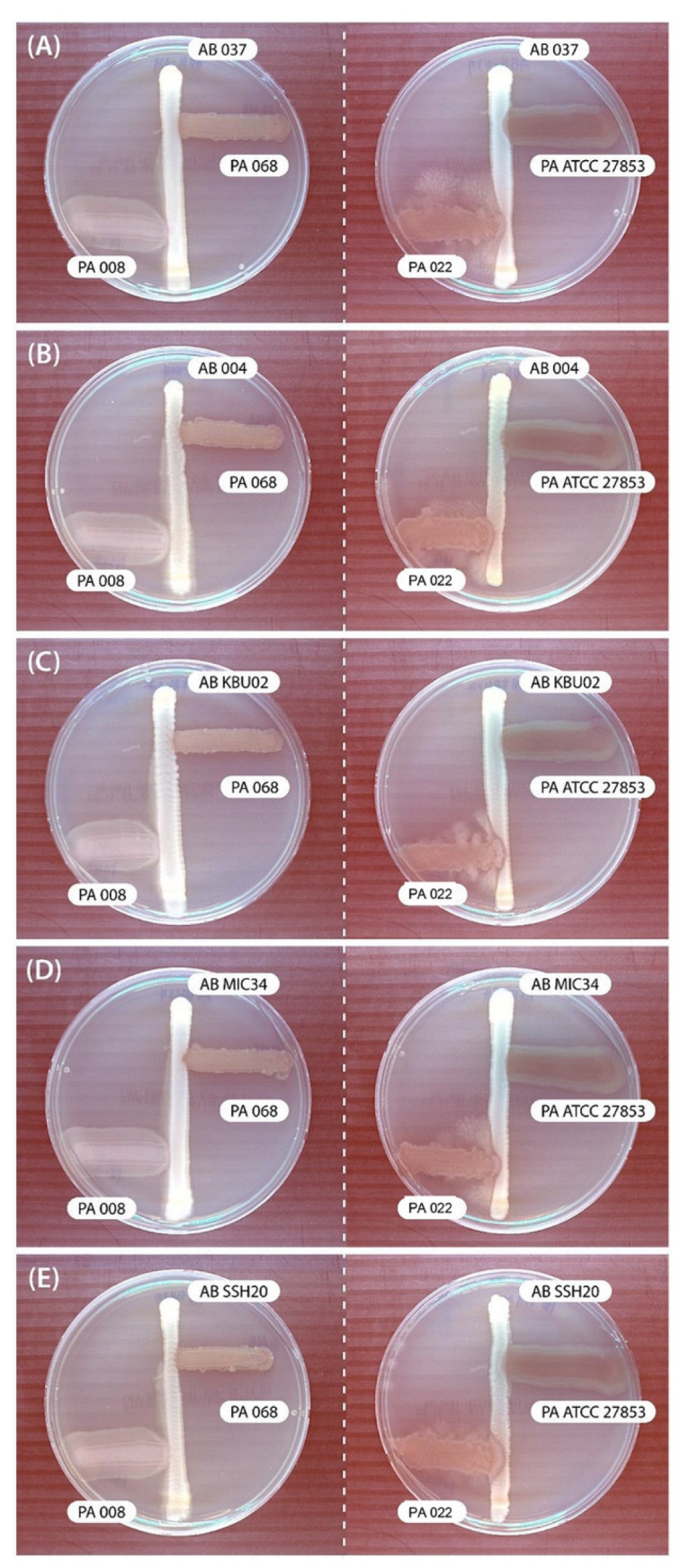
Results of modified cross-streak assay examining the effects of *P. aeruginosa* (PA) against *A. baumannii* (AB) after incubating for 72 h. (**A**) AB 037 vs. PA, (**B**) AB 004 vs. PA, (**C**) AB KBU 02 vs. PA, (**D**) AB MIC 34 vs. PA, and (**E**) AB SSH 20 vs. PA.

**Figure 2 biomedicines-12-00556-f002:**
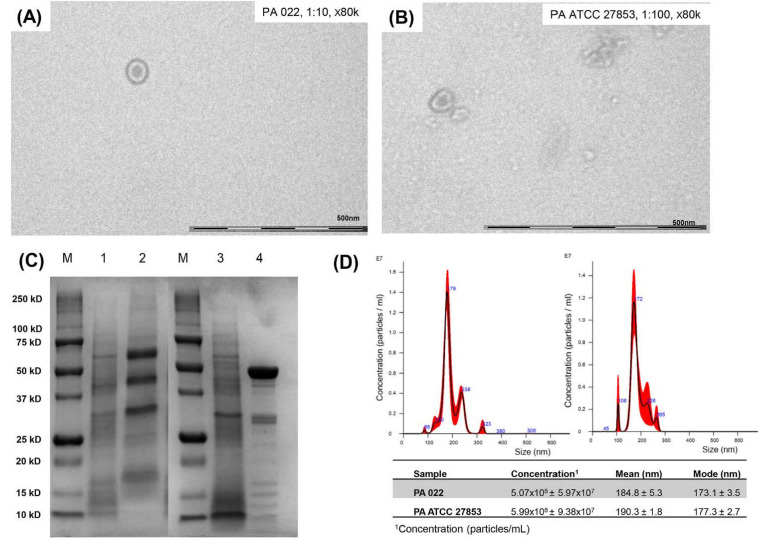
Results of OMV preparation in *P. aeruginosa* (PA 022 and PA ATCC 27853). (**A**) Transmission electron microscopy (PA 022), (**B**) transmission electron microscopy (PA ATCC 27853), (**C**) SDS-PAGE (M: protein molecular weight marker, 1: PA ATCC 27853 lysates, 2: PA ATCC 27853 OMVs, 3: PA 022 lysates, and 4: PA 022 OMVs), (**D**) nanoparticle tracking assay (right: PA 022 OMVs, left: PA ATCC 27853 OMVs).

**Figure 3 biomedicines-12-00556-f003:**
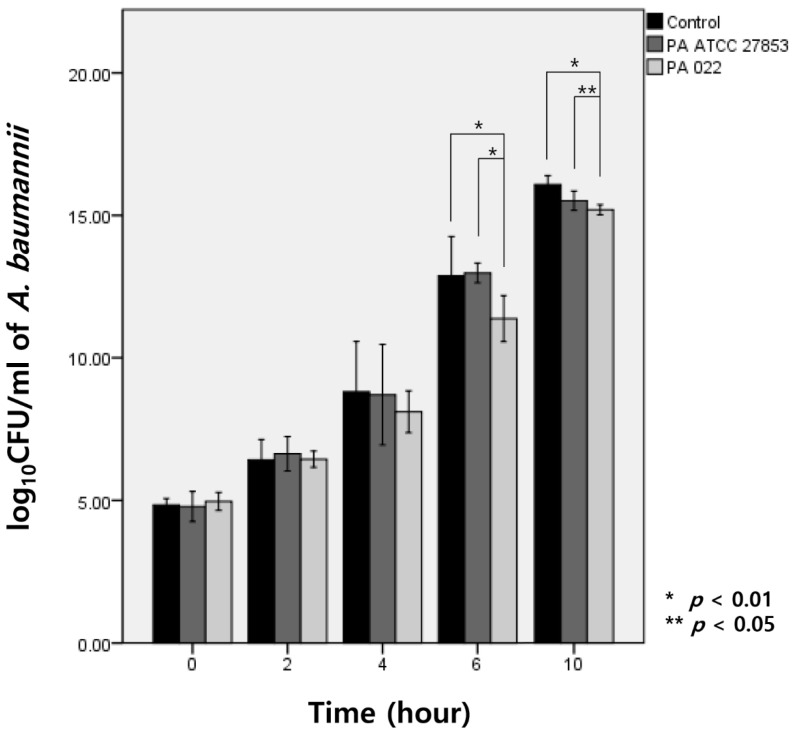
Modified time-kill assay of *A. baumannii* by *P. aeruginosa* (PA) OMVs, expressed as log CFU/mL of *A. baumannii* (AB).

**Figure 4 biomedicines-12-00556-f004:**
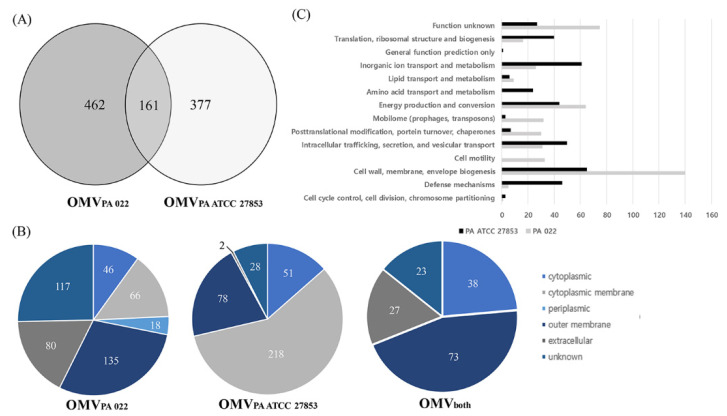
Identification of *P. aeruginosa* (PA 022 and PA ATCC 27853) OMV proteins. (**A**) Venn diagram shows the number of identified proteins from the PA 022 and PA ATCC 27853 OMVs. (**B**) Classification of subcellular locations of the identified proteins based on the number of proteins. (**C**) Functional classification of the PA 022 and PA ATCC 27853 OMV proteins.

**Table 1 biomedicines-12-00556-t001:** Outer membrane and extracellular proteins identified from *P. aeruginosa* OMVs.

UniProt Accession	Protein Name	No.	Average Mass (kDa)	Gene Name	Subcellular Location
PA ATCC 27853 OMVs					
A0A8D9JJ04	Outer membrane protein assembly complex protein	2	87,678.00	yaeT	Outer membrane
A0A246FDS1	Peptidoglycan-associated protein	14	17,963.58	pal	Outer membrane
A0A069PYF8	Lipid A deacylase	4	18,399.60	pagL	Outer membrane
A0A8D9JLC9	Lipid A 3-O-deacylase	3	18,394.00	pagL	Outer membrane
A0A8H2EWC8	Multidrug efflux RND transporter outer membrane protein	24	51,163.11	oprN	Outer membrane
I1SY51	Peptidoglycan-associated lipoprotein	7	16,841.40	oprL	Outer membrane
A0A8G3E6L1	Transport protein	3	45,130.33	IPC1518_14570	Outer membrane
A0A0C6ELC4	Long-chain fatty acid transport protein	3	45,566.67	fadL1	Outer membrane
A0A8G2ZED3	Outer membrane protein assembly factor protein	11	88,435.73	bamA	Outer membrane
A0A8G5T8N0	TolC family protein	1	47,611.00	IPC183_04225	Outer membrane
A0A2R3J2M9	Efflux transporter outer membrane factor lipo NodT family protein	1	51,290.00	CSB93_4029	Outer membrane
A0A241XQD4	RND transport protein	1	51,161.00	CAZ10_13505	outer membrane
A0A069Q437	Outer membrane protein transport protein	3	45,562.00	ALP65_01642	Outer membrane
A0A0A8RDW0	Porin D	1	46,900	oprD3	Outer membrane
A0A8G6ZVZ6	Phage tail protein	2	72,190.00	IPC65_03235	Extracellular
PA 022 OMVs					
A0A127MNT2	Glycine zipper 2TM domain-containing protein	7	15,660.67	slyB_1	Outer membrane
A0A1C7BTT8	Rick_17kDa_Anti domain-containing protein	5	15,649.00	Q058_03352	Outer membrane
A0A1C7BQ09	TonB-dependent copper receptor	26	79,249.31	Q058_00426	Outer membrane
A6VD89	Outer membrane efflux protein	4	53,298.00	PSPA7_5705	Outer membrane
A6V9E3	Lipoprotein	5	14,798.00	PSPA7_4325	Outer membrane
A0A1C7BA19	Lipid A deacylase	7	18,403.86	pagL	Outer membrane
Q51487	Outer membrane protein OprM	28	52,607.55	oprM	Outer membrane
V6ALH3	Outer membrane protein oprJ	12	51,955.83	oprJ	Outer membrane
V5UUX5	Outer membrane porin protein OprD	10	47,550.67	oprD	Outer membrane
A0A8D9JM75	Copper transport outer membrane porin OprC	5	78,644.00	oprC	Outer membrane
G3XCK3	Channel protein TolC	21	53,345.61	opmH	Outer membrane
A0A8G3VVE4	YgdI/YgdR family lipoprotein	2	8095.00	IPC1338_27820	Outer membrane
A0A072ZKU5	17 kDa surface antigen	1	15,649.00	slyB	Outer membrane
A6V457	Putative outer membrane protein	1	46,879.00	PSPA7_2476	Outer membrane
A0A8G1KGY7	YadA-like family protein	1	90,268.00	K3T18_31340	Outer membrane
A0A1B1SN37	Pilin	3	15,958.33	pilA	Extracellular
V6AAB7	Elastase	19	51,771.00	lasB	Extracellular
A7LI11	Peptidase M4 family protein	10	53,672.10	IPC113_02740	Extracellular
Q9KW03	Phage tail protein	17	68,141.85	hr1	Extracellular
A6V997	Flagellin	30	31,929.25	fliC	Extracellular
A0A8G3YS22	Pseudolysin	1	53,687.00	Q041_00565	Extracellular

## Data Availability

The datasets generated or analyzed during the current study are available from the corresponding author upon reasonable request.

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
