# Peer review of "Characterization of the Outer Membrane Vesicles of Pseudomonas aeruginosa Exhibiting Growth Inhibition against Acinetobacter baumannii"

_biomedicines, 2024, doi:10.3390/biomedicines12030556_

Round 1

Reviewer 1 Report

Comments and Suggestions for Authors

Characterization of the outer membrane vesicles of Pseudomonas aeruginosa exhibiting growth inhibition against Acinetobacter baumannii by Jin Woong Suh &, Jae Seong Kang, Jeong Yeon Kim. Sun Bean Kim, Young Kyung Yoon and Jang Wook Sohn

This study elucidates the properties of Pseudomonas aeruginosa outer membrane vesicles (OMVs) and their impact on Acinetobacter baumannii in vitro. Our research highlights the suppressive effect of PA022 OMVs on A. baumannii proliferation, illuminating their pivotal role in modulating bacterial dynamics. The proteomic analysis revealed the presence of proteins linked to virulence, motility, and post-translational modifications (PTM), laying the groundwork for subsequent research focused on unraveling the pathogenic capabilities of P. aeruginosa OMVs. These findings represent a fundamental step towards exploiting OMVs' therapeutic potential against bacterial infections, thereby enriching our comprehension of microbial interactions.

The work is very interesting and further investigations are a must with a more quantitative description of the inhibitory effect in a liquid medium where all parameters of growth inhibition will be measured.  

How adequate is an expression of OMVs as log CFU/mL of A. baumannii.

Figure 3 mentions A and B segments but this is not clearly explained, error? This is the most interesting figure, should be well explained, and match the agar cross-streak assay results. If OMVs are used – is it possible that they lose activity, and the inhibition effect might be even bigger? Why there is no liquid culture competition of live organisms, that can be simply recorded with qPCR?

Author Response

Biomedicines (Ref. No. biomedicines-2871992)

Title of Paper: Characterization of the outer membrane vesicles of Pseudomonas aeruginosa exhibiting growth inhibition against Acinetobacter baumannii

21 Feb 2024

Dear Editor

We appreciate the time and effort taken by the editor and referees in reviewing our manuscript.

We have revised the manuscript based on the reviewers’ valuable comments. We have addressed all the issues indicated in the review report and listed the specific changes made in the revised manuscript.

We hope that these changes will make our manuscript suitable for publication in your esteemed journal.

Sincerely yours,

Jang Wook Sohn, MD, PhD

Division of Infectious Diseases, Department of Internal Medicine,

Korea University Anam Hospital,

Korea University College of Medicine,

73, Inchon-ro, Seongbuk-gu, Seoul 02841, Republic of Korea.

Tel: +82-2-920-5341

Fax: +82-2-920-5616

Email: jwsohn@korea.ac.kr

List of changes

Reviewer(s)’ Comments to Author

Reviewer #1

1) How adequate is an expression of OMVs as log CFU/mL of A. baumannii.

Answer: Thank you for your comment. In our research, we aimed to confirm whether OMVs of P. aeruginosa inhibit the growth of A. baumannii using a modified time kill assay. Previous studies on growth inhibition by OMVs have traditionally confirmed changes in log CFU/mL of target bacteria (1,2). Therefore, we adopted this research method for our experiments. Additionally, after observing that the growth of A. baumannii was inhibited by OMVs from PA 022 compared to standard PA strains, we believe that expressing the results in Log CFU/mL is appropriate as a method to confirm the inhibitory effect of OMVs on bacterial growth. However, evaluation through other methods such as quantitative PCR (qPCR) will also be necessary, and this will be reflected in future follow-up studies.

  1. Wang Y, Hoffmann JP, Baker SM, Bentrup KHZ, Wimley WC, Fuselier JA, Bitoun JP, Morici LA. Inhibition of Streptococcus mutans biofilms with bacterial-derived outer membrane vesicles. BMC Microbiol. 2021;21(1):234.
  2. Roszkowiak J, Jajor P, GuÅ‚a G, Gubernator J, Å»ak A, Drulis-Kawa Z, Augustyniak D. Interspecies Outer Membrane Vesicles (OMVs) Modulate the Sensitivity of Pathogenic Bacteria and Pathogenic Yeasts to Cationic Peptides and Serum Complement. Int J Mol Sci. 2019;20 (22):5577. 

2) Figure 3 mentions A and B segments but this is not clearly explained, error?  

Answer: Thank you for your feedback. As you mentioned, we have corrected A and B segments as errors.

Page 9, Fig 3. Modified time-kill assay of A. baumannii by P. aeruginosa (PA) OMVs, expressed as log CFU/mL of A. baumannii (AB).

3) This is the most interesting figure, should be well explained, and match the agar cross-streak assay results.

Answer: Thank you for your feedback. As you mentioned, we have elucidated the correlation between the results of cross-streak assay and time-kill assay in the discussion section.

Page 10, 4. Discussion. Both the cross-streak assay and the time-kill assay are well-known methods for evaluating the antimicrobial activity in vitro. The cross-streak assay is a rapid screening method for assessing microorganism antagonism, while the time kill assay is recognized as the most appropriate method for determining bactericidal effects (27). Our study revealed a significant inhibition zone of A. baumannii by PA022 was observed after 72 hours in the cross-streak assay, whereas PA ATCC 27853 formed distinct colonies overlapping with A. baumannii. Furthermore, the time-kill assay revealed significant growth inhibition of A. baumannii by extracted PA022 OMVs after 6 h, and PA ATCC 27853 also exhibited growth inhibition effects compared to the control group after 10 h. These findings suggest that the inhibition phenomenon formed distinct colony-like boundaries in the cross-streak assay.

Page 14, Reference

  1. Balouiri M, Sadiki M, Ibnsouda SK. Methods for in vitroevaluating antimicrobial activity: A review. J Pharm Anal. 2016;6(2):71-79.

4) If OMVs are used – is it possible that they lose activity, and the inhibition effect might be even bigger?

Answer: I am very grateful for your comment. We hypothesized that the mechanism by which OMVs inhibit bacterial growth is mediated by proteins exhibiting virulence, as identified through proteomic analysis. However, we also consider it important to compare changes in activity and the resulting differences in OMV effects. Nevertheless, in this study, our aim is to confirm whether P. aeruginosa OMVs inhibit A. baumannii growth and identify candidate protein groups that may demonstrate inhibitory potential through proteomic analysis. Therefore, we will investigate and explore the effects of changes in OMV activity through subsequent research for comparison and evaluation.

5) Why there is no liquid culture competition of live organisms, that can be simply recorded with qPCR?

Answer: I am very grateful for your feedback. We measured the colony count of A. baumannii in response to OMVs using the modified time-kill assay method, allowing us to assess changes in log CFU/mL. As you suggested, we believe that the use of quantitative PCR (qPCR), which demonstrates high sensitivity, rapid turnaround time, and is less labor-intensive, would be necessary. However, qPCR is limited in its ability to distinguish between DNA from live versus dead cells (1). Considering that OMVs contain not only proteins but also genetic materials such as DNA and RNA, comparing the expression levels between OMVs and target strains would not be straightforward. Since the purpose of this study was to confirm whether P. aeruginosa OMVs inhibit A. baumannii and to analyze potential virulent proteins through proteomic analysis, we will conduct additional studies comparing and analyzing colony count methods and qPCR using liquid culture when performing future research related to OMV-mediated inhibition of A. baumannii.

  1. Thilakarathna SH, Stokowski T, Chui L. An Improved Real-Time Viability PCR Assay to Detect Salmonellain a Culture-Independent Era. Int J Mol Sci. 2022;23(23):14708.

Reviewer 2 Report

Comments and Suggestions for Authors

The authors at the article entitled “Characterization of the outer membrane vesicles of Pseudomonas aeruginosa exhibiting growth inhibition against Acinetobacter baumannii“ revealed the interaction of Pseudomonas aeruginosa (PA) with Acinetobacter baumannii by examining the effect of PA outer membrane vesicles on the growth of Acinetobacter baumannii. The description of the technical side of the research is very extensive, and the actual discussion is too concise and addressed only to a narrow group of specialists, therefore the manuscript in its presented form should be sent to a trade journal. Moreover, the authors extensively listed the weaknesses of their research and the directions for further research. After performing these studies and/or after thoroughly editing the manuscript so that it is accessible to a wide range of readers of MDPI journals, it may be considered for publication in the journal Biomedicines. Since the conducted research has signs of usefulness for other researchers, in the event of a thorough revision of the manuscript and re-submission for consideration of suitability for publication in MDPI or favorable reviews from other reviewers, a few minor errors are listed below:

Page 1

At the 2 line of Introduction is … in vitro. … , but should be …in vitro. … . Comment: the phrase does not come from English, so it should be written in italics.

Page 2

At the end of Introduction paragraph is … in vitro. … , but should be …in vitro. … . Comment: the phrase does not come from English, so it should be written in italics. Similar errors are: on page 10, line 10 of paragraph 4. Discussion on page 12, line 15 and at line 2 of paragraph 5. Conclusion

Page 5

At paragraph 2.6. is … (range: 22.5℃–22.6℃). … , but should be better (range: 22.5–22.6℃). … . See such line 5 at paragraph 2.5.

Page 7

At first line is … (Fig1) … , but should be … (Fig 1) … . Comment: A space please add. See page 8 at lines 3 and 6.

At lines 2, 3, and 5 of paragraph 2.8.1. is … NH4HCO3 … , but should be … NH4HCO3 … . Comment: The IUPAC nomenclature recommended entering the number of elements/groups in a subscript.

At line 7 of paragraph 2.8.1. is abbreviation … TFA … , but for the needs of MDPI readers, please provide the full name of trifluoroacetic acid.

Page 8

At first line is … (Fig2A, B) … , but should be … (Fig 2A, B) … Comment: Please add a space. See page 8, lines 3 and 6.

Page 10

In the first paragraph there is a link to the website (https://www.ncbi.nlm.nih.gov/research/cog/). … which should be moved to the References section.c

Page 11

In lines 24 and 25 there is an abbreviation Opr … , but please add its full form once.

Author Response

Biomedicines (Ref. No. biomedicines-2871992)

Title of Paper: Characterization of the outer membrane vesicles of Pseudomonas aeruginosa exhibiting growth inhibition against Acinetobacter baumannii

21 Feb 2024

Dear Editor

We appreciate the time and effort taken by the editor and referees in reviewing our manuscript.

We have revised the manuscript based on the reviewers’ valuable comments. We have addressed all the issues indicated in the review report and listed the specific changes made in the revised manuscript.

We hope that these changes will make our manuscript suitable for publication in your esteemed journal.

Sincerely yours,

Jang Wook Sohn, MD, PhD

Division of Infectious Diseases, Department of Internal Medicine,

Korea University Anam Hospital,

Korea University College of Medicine,

73, Inchon-ro, Seongbuk-gu, Seoul 02841, Republic of Korea.

Tel: +82-2-920-5341

Fax: +82-2-920-5616

Email: jwsohn@korea.ac.kr

List of changes

Reviewer(s)’ Comments to Author

Reviewer #2

1)

Page 1. At the 2 line of Introduction is … in vitro. … , but should be …in vitro. … . Comment: the phrase does not come from English, so it should be written in italics. Page 2. At the end of Introduction paragraph is … in vitro. … , but should be …in vitro. … . Comment: the phrase does not come from English, so it should be written in italics. Similar errors are: on page 10, line 10 of paragraph 4. Discussion on page 12, line 15 and at line 2 of paragraph 5. Conclusion

Answer: As per your recommendation, we revised ‘in vitro’ to in vitro in italics.

Page 1, Abstract. Introduction: We investigated the Pseudomonas aeruginosa (PA) outer membrane vesicles (OMVs) and their effect on Acinetobacter baumannii (AB) growth in vitro.

Page 2, 1. Introduction. Therefore, this study aimed to investigate the interactions between P. aeruginosa and A. baumannii and determine the microbiological characteristics of P. aeruginosa OMVs against A. baumannii in vitro.

Page 10, 4. Discussion. Nevertheless, a previous study did not observe the antibacterial effects of P. aeruginosa extracellular compounds on A. baumannii during in vitro co-culture (26).

Page 12, 4. Discussion, limitation. Our study has several limitations. First, the study primarily focused on characterizing OMVs and their potential in vitro inhibitory effects….

Page 12, 5. Conclusion. This study revealed the characteristics of P. aeruginosa OMVs and their effects on A. baumannii in vitro.

2)

Page 5. At paragraph 2.6. is … (range: 22.5℃–22.6℃). … , but should be better … (range: 22.5–22.6℃). … . See such line 5 at paragraph 2.5.

Answer: As per your recommendation, we revised ‘22.5℃–22.6℃’ to 22.5–22.6℃.

Page 5, paragraph 2.6. The measurements were performed at an ambient room temperature (range: 22.5–22.6℃).

3)

Page 7. At first line is … (Fig1) … , but should be … (Fig 1) … . Comment: A space please add. See page 8 at lines 3 and 6.

Answer: As per your recommendation, we revised ‘Fig1’ to Fig 1.

Page 7, paragraph 3.1. PA022 showed significant growth inhibitory zones for all tested A. baumannii adjacent to P. aeruginosa after 72 h (Fig 1).

4)

At lines 2, 3, and 5 of paragraph 2.8.1. is … NH4HCO3 … , but should be … NH4HCO3 … . Comment: The IUPAC nomenclature recommended entering the number of elements/groups in a subscript.

Answer: As per your recommendation, we revised ‘NH4HCO3’ to NH4HCO3.

Page 6, paragraph 2.8.1. Gels were washed with 5% acetonitrile (ACN) (Sigma-Aldrich, St. Louis, MO, USA) in 25 mM NH4HCO3 (Sigma-Aldrich, St. Louis, MO). After cutting the gels into small pieces, they were destained twice with 50% ACN in 25 mM NH4HCO3, dehydrated with ACN, and dried. Proteins were enzymatically digested overnight at 37 °C using trypsin (Promega, Madison, WI, USA) in 40 mM NH4HCO3.

5)

At line 7 of paragraph 2.8.1. is abbreviation … TFA … , but for the needs of MDPI readers, please provide the full name of trifluoroacetic acid.

Answer: As per your recommendation, we added the full name of TFA.

Page 6, paragraph 2.8.1. The resulting tryptic peptides were extracted twice with 0.5% trifluoroacetic acid (TFA)……

6)

Page 8. At first line is … (Fig2A, B) … , but should be … (Fig 2A, B) … Comment: Please add a space. See page 8, lines 3 and 6.

Answer: As per your recommendation, we revised ‘Fig2A, B’ to Fig 2A, B.

Page 8, paragraph 3.2. TEM analysis confirmed the presence of spherical phospholipid bilayer structures in PA022 OMVs and PA ATCC 27853 OMVs (Fig 2A, B).

7)

Page 10. In the first paragraph there is a link to the website … (https://www.ncbi.nlm.nih.gov/research/cog/). … which should be moved to the References section.

Answer: As per your recommendation, we moved the website (http://ncbi.nlm.nih.gov/research/cog/) to reference section as reference 23.

Page 10, paragraph 3.4

Page 14, reference 23

  1. Database of Clusters of Orthologous Genes (COGs). Available online: https://www.ncbi.nlm.nih.gov/research/cog/ (accessed on 20 01 2024)

8)

Page 11. In lines 24 and 25 there is an abbreviation … Opr … , but please add its full form once.

Answer: As per your recommendation, we added the full name of Opr.

Page 11, 4. Discussion. Octatricopeptide repeat (Opr) family was identified….

Round 2

Reviewer 2 Report

Comments and Suggestions for Authors

The authors submitted a revised version of their manuscript titled “Characterization of the outer membrane vesicles of Pseudomonas aeruginosa exhibiting growth inhibition against Acinetobacter baumannii“ for review. Indeed, the authors investigated the effect of Pseudomonas aeruginosa (PA) outer membrane vesicles (OMVs) on the growth of Acinetobacter baumannii (AB) in vitro, with success. Moreover, in the last paragraph of the discussion chapter, the authors focused on a detailed demonstration of the limitations of the research and set goals for further research, which strongly and incorrectly suggests that the proposed type of scientific work is rather a dissertation. Please summarize this text and redraft it into the desired form of an article. These research goals, which are in fact shortcomings of which the authors are aware, should be the subject of the authors' subsequent publications, and not a research suggestion for the reader.

After a thorough correction of the nature of the dissertation into the desired article and removal of the shortcomings listed below, the manuscript can be sent to further stages of production for publication in the journal "Biomedicines":

Throughout the manuscript, the style of providing references should be adapted to the standards of the journal/publisher, see for example:

https://doi.org/10.3390/biomedicines12030507

Abstract:

Is … similiar sizes … . Comment: Please check if it is correct and correct it if necessary.

The abstract subsection "Introduction:" is bold, but the subsections "Material and methods:", "Results:", and "Conclusion:" are not bold. Comment: Please standardize, or alternatively remove the subsection names as they are completely unnecessary.

Introduction:

As indicated above, at page 1 the link style for example is: ... cells. (1). ... , … transport. (2). …, … Microorganisms. (3). Additionally, …, …  enzymes. (4). …, … pathogenesis. (5). Furthermore, …, and should be respectively ... cells [1]. ... , … transport [2]. …, … Microorganisms [3]. Additionally, …, …  enzymes [4]. …, … pathogenesis [5]. Furthermore, …, etc. Comment: Please be sure to revise the style of references throughout the manuscript.

At page 2 is: … a VITEK II (BioMérieux, Hazelwood, MO, USA) and a MicroScan WalkAway-96 Plus system (Beckman Coulter, Inc., CA, USA) (Table 1). The … , but should be … a VITEK II (BioMérieux, Hazelwood, MO, USA) and a MicroScan WalkAway-96 Plus system (Beckman Coulter, Inc., CA, USA) (Table 1). The … . Please standardize the font size.

On page 3, table 1; column 3 from the left, and last column (first from the right), please increase their sizes (short words and abbreviations should fit in one column cell).

The authors wrote the titles of Figures 2 and 4 in bold, in other cases in normal style (Figures 1, 3). Please standardize.

At 2.8.1., page 6 is with space … 37 °C … , but above (at page 5, on 2.6.) the space is lack ... 22.6℃ … . Comment: Please standardize throughout the manuscript.

The Abstract is not part of the manuscript, so the Conclusion section 5 should be more detailed. Comment: Please expand on the conclusions, because the research is quite detailed and deserves a slightly broader summary, moreover, readers study this element before possibly studying the entire manuscript.
